# Energy Harvesting and Water Saving in Arid Regions via Solar PV Accommodation in Irrigation Canals

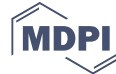

**Ayman Alhejji [1],\*, Alban Kuriqi [2], Jakub Jurasz [3] and Farag K. Abo-Elyousr [4]**

1 Department of Electrical and Electronics Engineering Technology, Yanbu Industrial College, Yanbu Industrial City 41912, Saudi Arabia

2 CERIS, Instituto Superior Técnico, Universidade de Lisboa, 1049-001 Lisbon, Portugal; alban.kuriqi@tecnico.ulisboa.pt

3 Faculty of Environmental Engineering, Wroclaw University of Science and Technology, 50-370 Wroclaw, Poland; jakubkamiljurasz@gmail.com

4 Electrical Engineering Department, Faculty of Engineering, Assiut University, Assiut 71516, Egypt; farag@aun.edu.eg

\* Correspondence: alhejjia@rcyci.edu.sa

**Abstract:** The Egyptian irrigation system depends mainly on canals that take water from the River Nile; nevertheless, the arid climate that dominates most of the country influences the high rate of water losses, mainly through evaporation. Thus, the main objective of this study is to develop a practical approach that helps to accommodate solar photovoltaic (PV) panels over irrigation canals to reduce the water evaporation rate. Meanwhile, a solar PV panel can contribute effectively and economically to an on-grid system by generating a considerable amount of electricity. A hybrid system includes a solar PV panel and a diesel generator. Several factors such as the levelized cost of energy (LCOE), total net present cost, loss of power supply probability, and greenhouse gas emissions should be considered while developing a technoeconomically feasible grid-connected renewable integrated system. A mathematical formulation for the water loss was introduced and the evaporation loss was monthly estimated. Thus, this study also aims to enhance an innovative metaheuristic algorithm based on a cuckoo search optimizer to show the way forward for developing a technoeconomic study of an irrigation system integrated with an on-grid solar PV panel designed for a 20-year lifespan. The results are compared using the mature genetic algorithm and particle swarm optimization to delimit the optimal size and configuration of the on-grid system. The optimal technoeconomic feasibility is connected to the graphical information system to delimit the optimal length and direction of the solar PV accommodation covering the canals. Finally, based on the simulated results, the optimal sizing and configuration of the irrigation-system-integrated on-grid solar PV accommodation have less impact on the LCOE without violating any constraint and, at the same time, generating clean energy.

**Keywords:** cuckoo search; economic feasibility; microgrids; renewable energy; water evaporation

## 1. Introduction

Egypt's agricultural land area is about 8.5 million feddans [1,2], or about 3.5% of Egypt's total area, a small area compared to its nutritional needs. These statistical numbers show the thinking on land reclamation, which needs additional quantities of water. Climate change significantly influences the spatio-temporal circulation of precipitation and evaporation, impacting water availability, especially in arid and semiarid regions such as Egypt [3–5]. Overall, Egypt's agricultural activities consume more than 80% of the available quantity of water withdrawn from the River Nile; moreover, with the Grand Ethiopian Renaissance Dam's construction, water scarcity is expected to be a severe issue in Egypt [6,7]. Thus, reducing water losses in irrigation networks, such as leachate and evaporation, might save remarkable amounts of water [8]. Leachate losses are usually



reduced by lining up canals. While, evaporation reduction can be achieved by reducing the direct contact of the water bodies with the atmosphere, which requires finding adaptation measures for the existing canals and developing sustainable design concepts for future irrigation canals. Therefore, it is essential to find alternative sustainable solutions to reduce water losses due to evaporation and, at the same time, generate renewable energy, which would contribute to the reduction of greenhouse-gas emissions. In this regard, the canal-top photovoltaic (PV) system is an emerging technology with rapidly increasing interest and application in practice [9–12]. Previous studies have found that this system has lower installment costs and higher efficiency than land-based solar systems and saves productive land areas that can be used for other purposes instead of being covered by PV panels [12]. It also contributes to saving a significant amount of water by reducing evaporation by nearly 40%. In the literature, several water-body-top PV plants have been reported [10,12–18]. Cazzaniga et al. [13] discuss several floating PV plants' designs and the corresponding performance evaluations on the lake, ponds, and water reservoirs. Nearly 30% of the energy increase is due to the cooling effect. Kumar, et al. [10] studied a 10 MW canal-top PV plant's performance and found that the PV's efficiency was ten times greater than land-based PV power plants. Santafé, et al. [14] investigated the construction features of floating PV for irrigation reservoirs to minimize water evaporation; they concluded that the PV accommodation was technically viable to reduce evaporation loss and generate electricity. Tajo-Segura canal's 5 MW-capacity PV system benefits were investigated by Colmenar-Santos, et al. [15], finding an annual savings of 226 k€ and corresponding loss reduction of 6.57 GWh. The wattage production of a floating and a land-based PV was considered in Yadav, et al. [16]. The experimental setup highlighted the increased efficiency of the floating-type PV compared to conventional systems. Kougias, et al. [17] studied the potential use of water dams to install PV plants. They concluded that the solar PV installation on dams' downstream sidebands could enhance unexploited hydropower potential. Sairam and Aravindhan [12] studied the canal-top solar panels' versus land-based panels' efficiency. The canal-top PV recorded 10% greater efficiency.

In Greece, the advantages of PV incorporation in irrigation in Thessaloniki-Imathia-Pella were introduced in Chrysochoidis-Antos and Chrysochoidis [19]. They recommended that incorporating PV could be enhanced by hybridization with wind turbines. The PV system was realized by developing the infrastructure associated with PV integration. In additional research conducted by Kougias, et al. [20], solar PVs' accommodation on the Mediterranean islands' existing dams was presented. So far, India has been a pioneering country in accommodating canal-top solar PV. A 10 kWp solar PV was installed over a 3.6 km portion of a canal [10]. This canal provides 16 hectares with 33,816 PV panels. The revenue generated by this project was estimated to be nearly 47.6% more than the same size land or ground-based PVs [21].

The majority of the water infrastructure in Egypt comprises small canals [6]. As the irrigation canals and climate conditions in Egypt have similarities to Indian canals and climate conditions, this gives us great motivation to propose a similar solution for Egyptian canals by accommodating canal-top PVs. The similarity with the Indian canals arises because the canals originate from rivers that constitute a natural water resource habitat in both countries. This makes canals' width depend on the irrigated areas [13,22]. However, other canals are used as a sink for water to return the surplus water to the Nile [1,2]. Many roads with electricity pylons can be found alongside these canals. Another motivation of this study is that almost all Egyptian canals are close to the primary grid. People live around canals, according to an official audit made into these communities [2]. Several decades ago, successive governments made much effort to connect countryside villages and related neighborhoods to the main grid, facilitating potential PV accommodations and construction works. This PV system also enables power production to residential customers or the primary grid.

Nevertheless, evaporation reduction is an important topic when investigating canals, rivers, and lakes. In Kumar, et al. [23], although the authors utilized reflectors to improve

canal-top PV efficiency in India, they demonstrated that a solar canal-top PV of 1 MW rating had saved nine million liters per year in one village and seven million liters in another. Some PV types demonstrated significant consequences due to water humidity. The different methods of evaporation estimations are considered in Meziani, et al. [24]. Bradei and Alsadeq [25] investigated covering Egyptian irrigating canals due to the evaporation levels in arid regions. Meziani, et al. [24] considered evaporation reduction in Algeria's arid areas from dam reservoirs. In AbdElrehim [26], a generic and straightforward overview of utilizing PV for water irrigation was introduced in Egypt. The optimization of reservoir water level and river flow in China was studied in Dai, et al. [27]. The investigation of the above works reveals an immediate need to consider the potential of a canal-top PV system to reduce the overall evaporation rates while at the same time generating clean electricity.

Generally speaking, PVs are installed on water bodies in two ways: floating and fixed water-mounted solar PVs. The PVs accommodated on reservoirs, ponds, and lakes usually are the floating type. In contrast, the PVs accommodated over canals and channels are the fixed type [10,23]. Kumar and Kumar [28] investigated the quantitative performance of the on-grid PV. A similar analysis was reported in Kumar and Kumar [22] in outdoor conditions. In additional research conducted by Perraki and Kounavis [29], climatic conditions, such as temperature and irradiance, on various PV types were explored. The qualitative performance of solar PVs, which relies on thermodynamics' 2nd law, was studied in Rawat, et al. [30]. A similar analysis to study energetic canal-top PV performance was investigated in Kumar and Kumar [31]. Singh, et al. [32] considered several techniques to investigate PV degradation. It was found that the I-V curves were the most appropriate technique. Kirmani, et al. [33] studied the reliability, performance, efficiency, and degradation at fixed intervals. The rate of degradation was recorded as being from 0.55 to 0.95 annually. Sharma and Chandel [8] investigated PV degradation in the western Himalaya weather-conditions zone with a degradation rate of 0.51% a year. The investigation of the above literature on water-body-based PVs reveals that most works have focused on mounting, reliability, degradation, and design. The existing findings of on-grid canal-top PV shows that economic feasibility analysis is rare in the literature, which is this study's other main goal.

Technoeconomic feasibility studies are a rich area in the literature. So far, important studies use ready-made software package, such as HOMER [34]. However, ready-made packages experience fixed and black-box entry for information. Other classic techniques such as graphical construction [35] and linear programming [36] were reported to solve several engineering issues. However, classic techniques might be captured within local optima. Alternatively, metaheuristic algorithms show the way forward towards solving technoeconomic feasibility problems. For example, a non-sorting genetic algorithm (NSGA) [37], ant colony [34], bee colony [38], harmony search [39], genetic algorithm [40], grey wolf optimizer [41], and crow search algorithm [42] were used to find a near-optimal solution for technoeconomic aspects. Despite the above work, there is still a need to develop advanced evolutionary algorithms. Besides, they might be stuck in local optima. Cuckoo search (CS) optimization has recently been used to find the global PV maximum point tracking as an online optimizer to change an electronic switch duty cycle [43]. It potentially demonstrated acceptable results compared to particle swarm optimization (PSO).

Moreover, CS optimization was used to reduce the costs to find the optimal size and configuration for a hybrid microgrid [44]. Furthermore, CS is straightforward and requires few parameters to proceed. It is also applicable to technoeconomic and power-system studies [44,45]. Although the literature is rich regarding employing various evolutionary algorithms to delimit the size and configuration of hybrid microgrid systems (HMG), few studies have considered canal-top PV technoeconomic issues to minimize the total power cost to the primary grid, thereby reducing global carbon emissions. Furthermore, there is a lack of studies considering evaporation reduction, especially in developing countries like Egypt. A near-optimal problem with the lowest levelized cost of energy (LCOE) was explored. This paper makes the following contributions to fill the gap in the

literature: (1) an accurate formulation of economic viability by determining the optimal size of on-grid canal-top PVs by minimizing the LCOE; (2) overcoming the economic feasibility analysis and proposing a new optimizer based on the CS algorithm; the obtained results are compared with the GA and PSO algorithms to validate the effectiveness of the developed CS algorithm; (3) a visual information system used to determine the required area and duration of the canal-top PV accommodated over the canal; (4) a generic analysis of evaporation conducted in a rural location in Assiut city, Egypt, to estimate water loss and suggest potential water saving through a canal-top PV system.

The remainder of this paper is organized as follow: Section 2 gives a brief description of the case study, problem statement, and a detailed explanation of the methodology; Section 3 shows the problem formulation, including main constraints, evaporation estimation, and the canal-top PV accommodation optimization; Section 4 presents the main results and discusses their relevance in comparison with existing literature. Finally, the main conclusions are given in Section 5.

## 2. Materials and Methods

### 2.1. Case Study

The Nile is the primary public irrigation network resource in Egypt. It flows from the Nile Basin countries to the south and heads north until it reaches the Mediterranean Sea. The river continues its course to the north until the al-Khayriya arches. It is then divided into two branches (Damietta in the east and Rasheed in the west). The two branches confine a wide cultivated land called the Delta. The second type, locally called Rayahs, is a watercourse less than the Nile but greater than a canal. Beyond the Delta, a barrage is located at four Rayahs: Nasser, Behera, Tawfiki, and Menoufia, as shown in Figure 1.

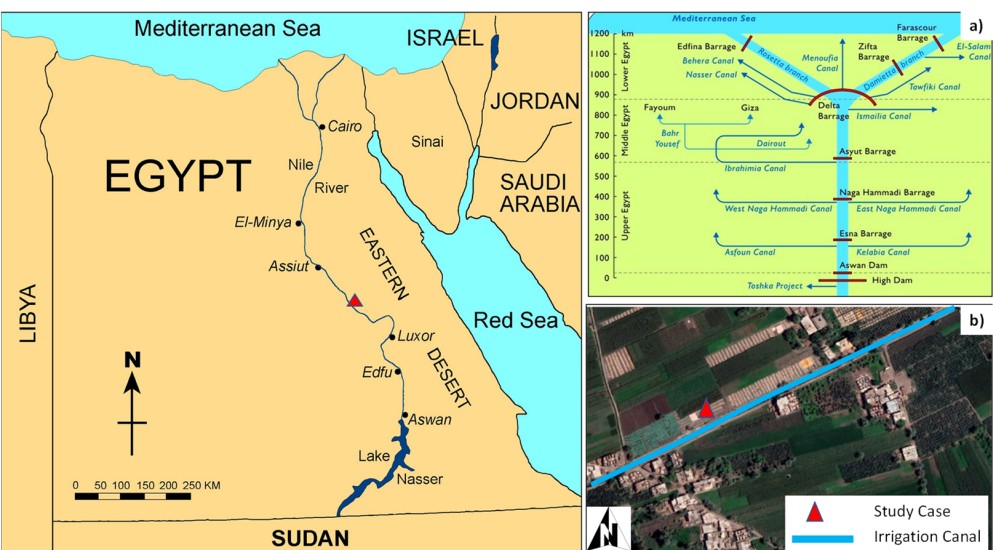

**Figure 1.** The Nile River flowing through Egyptian territory; (**a**) irrigation system in Egypt adapted after Abdrabbo [46] and (**b**) location of the case study site.

The third resource type is the main canals or the transferring canals, such as the Ibrahima canal in Asyut city, as in Figure 1a. Irrigation is merely carried out at the end. This type of watercourse takes the water directly from the Nile. The water is continuously flowing for about 11 months a year. The fourth type takes water from the main canals. However, small canals take their water from the fourth type in a weekly manner. The last type is made of small canals, entirely managed by the farmers themselves. A canal is considered sweet if used for irrigation and sour if it is a water sink, which returns the unused water to the River Nile.

## 2.2. Problem Statement

The current research concerns integrating canal-top PVs to provide electrical energy to the primary grid with minimal total costs and reasonable GHG emissions reduction. This study is conducted in the countryside of Assiut, Egypt, namely a village called Tasa, which is located at a latitude of 27.05° and longitude of 31.38°. The population of Tasa mainly depends on the agricultural economy and farming. The former comprises crop production and rearing livestock, while the latter is related to agriculture activities. This countryside village suffers from a severe low-voltage power supply, rendering illumination below the required level. During the daytime, canal-top PV accommodations might improve the lighting level from an electrical point-of-view due to the injected reactive power. The close-by low-voltage transformers and short transmission-line poles transfer the investigated site's electrical energy.

## 2.3. Canal-Top PV Accommodation

The current study is an ecotechnoeconomic feasibility workout in which canal-top PVs play a significant role. Figure 2 schematically demonstrates canal-top PV system accommodation over the canal and its benefits and depicts the on-grid solar PV system structure.

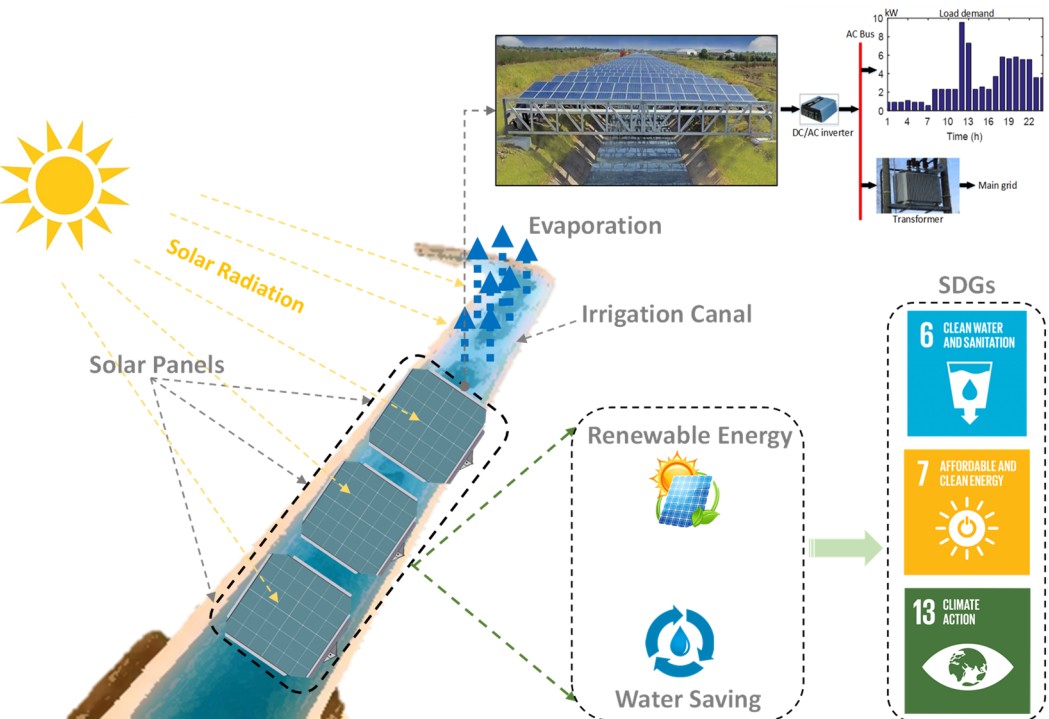

**Figure 2.** Structure of canal-top PV accommodation and its contribution to fulfilling SDGs..

The Sustainable Development Goals (SDGs) proposed almost 30 years ago by the United Nations represent the most relevant and ambitious global effort to advance the sustainable development of the global economy and well-being [47]. In this regard, it is believed that renewable energy sources and systems, such as canal-top PV systems, might contribute significantly, particularly in improving the following SDGs: ensuring the availability and sustainable management of water and sanitation for all (SDG 6), ensuring access to affordable, reliable, sustainable and modern energy for all (SDG 7), and taking urgent action to combat climate change and its impact (SDG 13) [48].

PV accommodation is directly connected to the DC bus. The load and the primary grid are connected to the AC bus. PV accommodation would potentially reduce the total evaporation rate of the canal-covered area. Meanwhile, it will also help reduce global GHG emissions due to shifting energy consumption away from the primary grid, which

mainly burns fossil fuels, to the canal-top PVs, which have a lower carbon footprint [17]. It is worth mentioning that the canals under study have no living fish due to the periodic nature of the water. Thus, canal-top PV accommodation is expected to not severely impact the ecological conditions of the water flowing through the canal because there are no living creatures, at least regarding this specific study case considered in this study. However, to ensure that good ecological conditions are maintained, continuously monitoring should be performed.

A steel frame, electrical energy metering equipment, and a concrete foundation are required whenever canal-top PVs are installed. Periodically, the water body flows for seven days and stops moving for another seven days. A significant and satisfactory solution for farmers is that there is no cultivated land conflict. The technical viewpoint is that canal-top PV accommodation will give a potential chance in the local network's reconfigurations, making local loads operate in isolated mode during the daytime. The economic feasibility analysis conducted in the current study paves the way forward for canal-top PV accommodation to continue with minimum LCOE and, hence, lower total present costs.

Consequently, the length of PV accommodation over the canal is determined according to the optimal PV rating estimation. As an outcome of the economic feasibility analysis, the power flow and the energy share are fixed economically. Hence, energy is purchased from the utility grid during off or weak irradiation. Surplus energy is sold to the primary grid. It is expected that, due to the cooling of the water body flow, at least during the non-humid months, that the PV panels would last longer than the ground-mounted solar PV panel's average lifetime of 25 years [10], while also generating more energy due to improved performance efficiency.

To tackle the technoeconomic feasibility analysis of canal-top PV accommodation, three key points are considered in this study as follows: (i) the mathematical formulation of on-grid canal-top PV accommodation of irrigation canals through minimizing the LCOE with satisfactory GHG-emissions reduction in addition to reasonable evaporation reduction, (ii) the mathematical modeling of the sole components and outlines of the developed optimizers, (iii) investigating the optimal tilt angle that offers maximum energy harvesting and reasonably meets the load demands.

### 2.4. System Modeling

The investigation includes electrical load, a DC/AC converter, and the primary grid in addition to the on-grid canal-top PV.

### 2.4.1. Canal-Top PV

The sun shines daily for 9 to 11 h at the investigated site, with a few cloudy days throughout the year. The power harvested by the PV module is given in (1), in which $S_{ref}$ (1000 W/m$^2$) and $T_{ref}$ (25 °C) are reference irradiation and temperature, respectively [34,37].

$$P_{PV}(t) = f_{PV} * P_{PVrate} * \frac{S(t)}{S_{ref}} \left[ 1 + \beta \left( T_s(t) - T_{ref} \right) \right] \tag{1}$$

where, $f_{PV}$ (0.98) is the derating factor, $\beta$ is the PV module temperature coefficient, $T_s(t)$ is the surface temperature at hour $t$, and $S$ is the corresponding solar irradiance. From now on, the key findings are the optimal PV module rating ($P_{PVrate}$) to minimize the LCOE and hence, total costs.

### 2.4.2. DC/AC Inverter

A power converter should afford the maximum load power ($P_l^m$). The power DC/AC inverter alters the DC solar PV power into AC, which is fed to the load. The DC/AC inverter rated power is:

$$P_{con} = \frac{P_l^m}{\eta_{inv}} \tag{2}$$

where, $\eta_{inv}$ is the DC/AC inverter efficiency.

### 2.4.3. Load Demand Profile

The load is chosen to be generic and suitable for such a countryside area. Most villages comprise a lump residential area with a few small surrounding inhabited areas. Despite being simple and straightforward, each small area is on-grid and includes ten houses. For this reason, the load consists of ten houses, a school, a primary healthcare center, and a worship house. The consumption is 80 kWh a day and a peak load of 9.5 kW [34]. The main grid is slack or floating bus where the surplus energy produced by the solar canal-top PV is sold, and any energy deficit is purchased.

### 2.5. Metaheuristic Optimizers

#### 2.5.1. Particle Swarm Optimization (PSO)

PSO is an optimizer in which birds and swarm foraging are simulated. Provided that all solutions and velocities are initialized randomly, it tries iteratively to optimize the current solution $(X_i^k)$ regarding the personnel solution (PB) and a global solution (GB), as in (3) and (4), respectively.

$$v_i^k = \omega_0 v_i^{k-1} + c_1 r_1 \left( GB - X_i^{k-1} \right) + c_2 r_2 \left( PB - X_i^{k-1} \right) \tag{3}$$

$$v_i^k = \omega_0 v_i^{k-1} + c_1 r_1 \left( GB - X_i^{k-1} \right) + c_2 r_2 \left( PB - X_i^{k-1} \right) \tag{4}$$

#### 2.5.2. Genetic Algorithm (GA)

GA is a mature optimizer in which the fitted genes or individuals are chosen for reproduction. The following steps are included in GA:

1.  The algorithm randomly starts reading the LCOE, and then an initial population (POPo) is assumed.
2.  The constraints in (POPo) are checked. The solutions that are outside the constraints are eliminated with a large penalty.
3.  The objective function (i.e., LCOE) is checked and evaluated at (POPo), and a new solution is generated (POPk).
4.  Another population is generated (POPk+1) via GA elitism, selection, crossover, and mutation.
5.  The constraints are checked at a solution (POPk+1). The simulation is stopped for a previously determined number of iterations. Ultimately, the best solution is printed.

#### 2.5.3. Cuckoo Search (CS)

CS is a population-based algorithm inspired by Yang [49] to imitate the cuckoo bird species' parasitic brood. There exist minimal parameters to be designed [50]. CS employs the Levy flight search technique, which makes it discover the search space's optimal values. To use the algorithm, the following procedures are satisfied [43,51]:

1.  The number of host nests is decided. Adult cuckoo birds choose random nests to lay the eggs. An adult cuckoo bird can lay an egg one at a time.
2.  Only eggs with better quality that the host parents cannot discover are moved to the upcoming generation.
3.  Provided that the probability (Pa) lies between 0 and 1, the host nest can delineate a cuckoo egg according to the Pa. Cuckoos' eggs discovered by the host birds are thrown away. Sometimes, the host bird might abandon the nest.

### 3. Problem Formulation

To obtain the on-grid canal-top PV's minimal operating costs, the developed optimizers are applied to find the minimum possible *LCOE* in (5) and (6), respectively. $T$ is the total hours set in a year.

$$J = min(f_1) \tag{5}$$

$$f_1(t) = \sum_{t=1}^{T} LCOE(PV_t) \qquad \forall \quad t\epsilon T \tag{6}$$

where $P_{purch}(t)$ is the purchasing power at hour $t$, and $C_{purch}(t)$ is the corresponding costs. LCOE is given in (7), in which the designation $C_t$ is the total yearly costs, and $E_l$ is the total yearly demand in kWh.

$$f_1(t) = \frac{C_t}{E_t} \tag{7}$$

The total annual costs are given in (8), in which $C_{int}$ is the total initial or CAPEX costs, $C_{OM}$ is the operational and maintenance costs, and $C_{rep}$ is the replacement costs.

$$C_t = C_{ini}\,CRF + C_{OM}\,PWF + + P_{purch}(t)\,C_{purch} \tag{8}$$

where,

$$CRF = \frac{i(1+1)^N}{(1+i)^{N-1}} \tag{9}$$

$$RF = \left(\frac{1+i_f}{1+i}\right)^N \tag{10}$$

$$PWF = RF\left(\frac{1-RF^N}{1-RF}\right) \tag{11}$$

where $N$ is the project lifetime (i.e., 20 years), provided that the interest rate $(i)$ and the inflation rates are known, the total net present costs (TNPC) are:

$$TNPC = \frac{C_t}{CRF} \tag{12}$$

The total initial coats are given in (13), where $C_{PV}$ is canal-top solar PV initial costs.

$$C_{ini} = P_{PVrate}\,C_{PV} \tag{13}$$

Additionally, the costs of service and maintenance ($C_{OM}$) are estimated to be 5% of the overall initial cost estimated in (13).

#### 3.1. GHG Emissions

GHG emissions are estimated based on the carbon footprint of the canal-top PV ($ft_{PV}$) and the grid ($ft_{grid}$), which is assumed to be fixed over time [34], as in (14).

$$GHG = \sum_{t=1}^{T} \left(ft_{PV}\,P_{PV}(t) + ft_{grid}P_{purch}(t)\right) \tag{14}$$

#### 3.2. Loss of Power Supply Probability (LPSP)

LPSP is a metric of system reliability as it is estimated depending on the unmet load at hour $t$ in terms of the load ($E_l$) and the total generation ($E_g$). Thus, the smaller the LPSP, the better system reliability is achieved.

$$LPSP = \frac{\sum_{t=1}^{8640}\left(E_l(t) - E_g(t)\right)}{\sum_{t=1}^{8640} E_l(t)} \tag{15}$$

### 3.3. Constraints

The solar PV's boundary capacity rating should be most significant to meet the load demand to ensure the on-grid canal-top PV's economic operation with satisfactory GHG emissions. The governing equation of the system constraints is given in [23]. Additionally, the boundaries of the canal to solar PV are set to 45 kW.

$$\sum_{t=1}^{8640} \left( P_{PV}(t) + P_{purch}(t) \right) \geq \sum_{t=1}^{8640} \frac{E_l(t)}{\eta_{inv}} \tag{16}$$

### 3.4. Evaporation Estimation

Evaporation represents one of the primary sources of water losses from open water bodies such as irrigation canals, especially in semiarid regions like Egypt [9]. The evaporation rate depends on several factors, such as atmospheric pressure and other meteorological variables, and water quality, among others [23]. In general, daily global horizontal irradiation (GHI), daily diffused horizontal irradiance (DHI), and notably, air temperature, relative humidity, and wind speed are among the primary meteorological variables that most influence the evaporation rate [23]. Table 1 shows the average values of the variables mentioned above; detailed information about each meteorological variable and also evaporation is given in Appendix A.

**Table 1.** Monthly statistics of the meteorological variables near the study case.

| Month | Avg. GHI [Wh/m$^2$] | Avg. DHI [W/m$^2$] | Avg. Temp. (°C) | Avg. $R_h$ (%) | Avg. Wind Speed (m/s) | Avg. $E$ (mm) |
|---|---|---|---|---|---|---|
| Jan | 4003.6 | 6.91 | 11.4 | 38.5 | 2.6 | 3.6 |
| Feb | 4905.1 | 10.4 | 13.7 | 40.8 | 3.1 | 4.4 |
| Mar | 6189.4 | 11 | 16.8 | 31.2 | 3.8 | 6.3 |
| Apr | 6684.4 | 14.7 | 21.9 | 25.6 | 3.9 | 8.7 |
| May | 7482.4 | 20.7 | 29.6 | 15.2 | 3.5 | 12.5 |
| Jun | 7515.3 | 27.8 | 32.2 | 21.6 | 4.2 | 14.9 |
| Jul | 7637.2 | 29.8 | 32.2 | 23.2 | 4.7 | 15.9 |
| Aug | 7174.5 | 29.8 | 31.9 | 24.4 | 4.0 | 14.0 |
| Sep | 6461.1 | 25.9 | 28.6 | 31.0 | 4.8 | 12.7 |
| Oct | 5351.7 | 20.8 | 26.3 | 32.0 | 3.8 | 9.9 |
| Nov | 4477.5 | 16.6 | 21.0 | 37.1 | 2.9 | 6.1 |
| Dec | 3979.2 | 10.2 | 14.1 | 50.8 | 3.3 | 4.0 |

Canal-top PV systems reduce evaporation by blocking direct solar radiation falling on water [10]. Depending on the data availability, the evaporation loss can be estimated using empirical formulas and methods [52]. In this study, evaporation was estimated using the Boutoutaou method [24], calibrated for humid/semihumid and arid/semiarid regions. The Boutoutaou estimates evaporation considering air temperature, relative humidity, and wind speed, measured from 2 to 10 m altitudes, using the following equation, Equation (17):

$$E = 0.403 \, n \, D^{0.73}(1 + 0.39V) \tag{17}$$

where, $E$ is the open-water evaporation (mm), in this case from the irrigation canals, $n$ is the number of days of the month, in this study the evaporation was estimated at daily scale and then aggregated to monthly scale, thus, $n = 1$, $V$ is the wind speed (m/s), and $D$ is the air saturation deficit (mb), computed by the following equation, Equation (18):

$$D = 0.0632 \, (100 - R_h)e^{0.0632(T_a)} \tag{18}$$

where, $R_h$ is the relative humidity (%), and $T_a$ is the air temperature (°C). The evaporation was then converted to water loss per unit area (i.e., per 1 m$^2$). Water loss variability ($W_L$) % was estimated at a daily time scale using Equation (19):

$$W_L = (E * A) * 100\% \qquad (19)$$

where $A$ is the area of the canal water body. The total water loss was also estimated at the monthly time scale

### 3.5. Optimizers Implementation

The grid-tied canal-top solar PV's optimal accommodation involves economic, environmental, and evaporation-reduction aspects. Herein, the developed optimizers pave the way towards the canal-top PVs' corresponding size. Figure 3 shows the developed CS optimizer's flow chart.

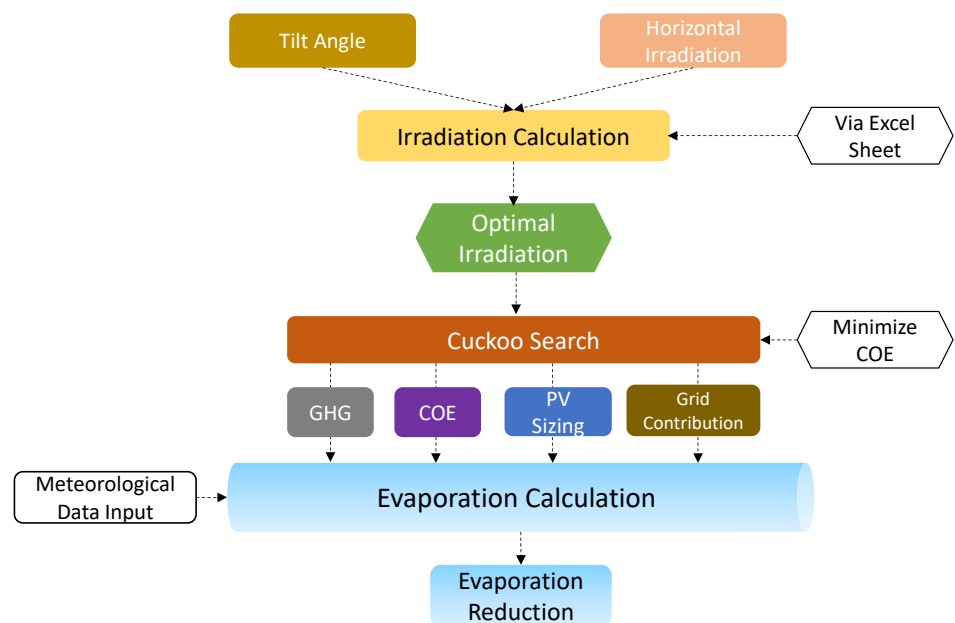

**Figure 3.** Canal-top PV accommodation sizing and impacts.

The algorithm starts reading the irradiation data and determines the optimal tilt angle. The CS works to find satisfactory LCOE, GHG emissions, and the economic proportion purchased from the primary grid. Figure 4 demonstrates the procedures of the CS optimizers.

The CS implementation starts reading the meteorological data followed by assuming a random value of the canal-top PV via the host nests to minimize the LCOE. All initial parameters of the optimizers are given in Appendix B. The CS works to minimize the LCOE by modifying the initial solution guessing via the Levy flight update technique, which ultimately results in the global optimal sizing of the canal-top PV. Contrarily, canal-top PV might have intermittent intervals to provide power. Besides, the obtained size has limited capacity, making it unable to respond to sudden load variations. The on-grid PV panel may experience a discontinuity. As shown in Figure 4, such energy breaks are covered by the developed energy management system by comparing the PV energy harvest and the load demand. The algorithm proceeds to send the sold energy to the primary grid. Otherwise, energy is purchased from the primary grid to meet the load demands. Developing an energy-management strategy is crucial to achieving the lowest economic and satisfactory environmental aspects and evaporation-reduction solutions. The flowing procedures were considered while performing the developed optimizers:

1.  For satisfactory evaporation reduction and environmental improvement, the PVs have a higher priority than the primary grid to meet the load demands,
2.  Canal-top PVs can provide the load demands; consequently, the excess energy is sold to the primary grid, as demonstrated in Figure 4, and
3.  Canal-top PVs cannot meet load demands. The deficit load energy is purchased from the primary grid.

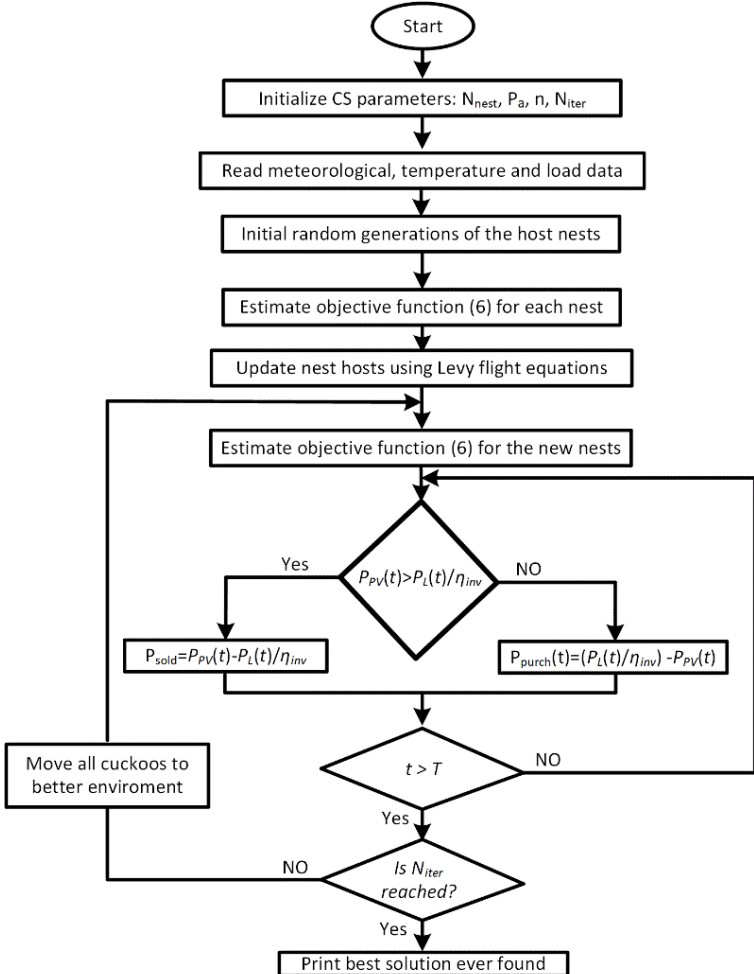

**Figure 4.** Canal-top PV accommodation flow chart via artificial cuckoo search.

## 4. Results and Discussion

### 4.1. Water Loss Due to Evaporation

The canal-top PV system is an innovative emerging technology that offers twofold benefits: water-saving and generating clean energy [12,23]. The PVs accommodated over the canal water bodies avoid direct solar radiation, resulting in lower water temperature than naked water bodies. All other climatic parameters vital for maintaining the good quality of the water body are not significantly affected [10,17]. In this study, water loss was estimated for the entire year as the irrigation canal operates for 11 months. Figure 5 shows that water loss variability varies within a day and differs from month to month, mainly due to atmospheric conditions.

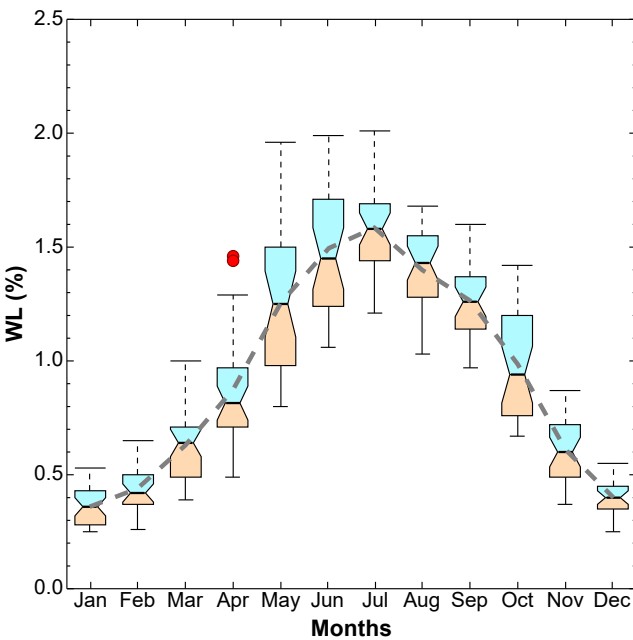

**Figure 5.** Notched boxplots show the daily variability of water loss for each month; the dashed line shows the mean value.

Overall, the winter season (i.e., December, January, and February) shows lower water loss than other seasons and is characterized by low variability. In contrast, other seasons, particularly summer (i.e., June, July, August), represent the highest water loss and higher variability. Higher variability in water loss during the summer season indicates that temperature and solar radiation have an essential role in the evaporation rate during the day and among seasons [24]. During the summer season, the total water amount lost due to evaporation is estimated between 40 and 50% (Figure 6).

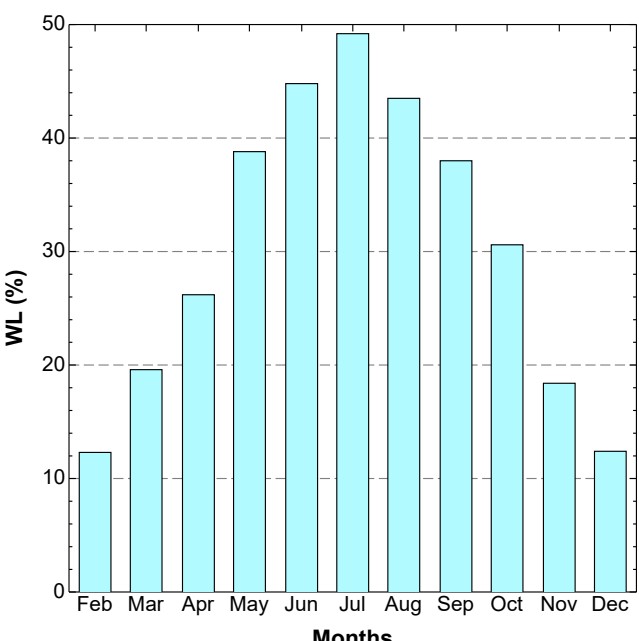

**Figure 6.** The total water loss for each month.

Thus, the accommodation of canal-top PV systems promotes a higher quantity of water-saving. In this study, due to resource limitation, it was impossible to conduct on-site or laboratory experiments to estimate precisely the water quantity that can be saved

through canal-top PV system accommodation. Nevertheless, considering the experimental findings from other studies conducted in regions characterized by climate and atmospheric conditions similar to ours, it is expected that canal-top PV system could save around 40% of the water otherwise lost due to evaporation from the irrigation canals, increase the lifespan of the solar panels, and generate clean energy, among other benefits [10,23].

### 4.2. Energy Production and Profitability

This study paves the way towards integrating solar PV systems to reduce the total amount of evaporation. Thus, the developed CS was applied to the on-grid canal-top PV to find optimal solar PV size with minimal total NPC by minimizing LCOE, considering the weather data of a countryside channel in Sahel Saleem city, Egypt [53]. The optimal solution confirms the economic and environmental benefits and evaporation reductions. Estimation of the canal-top PV was done using a discrete step increase of 0.1 kW. For the sake of safety, the maximum capacity constraint of the PV rating was set to 45 kW, which represents about three times the peak load demand. The Egyptian expenditures on the energy bill are shown in Figure 7, where beyond 1000 kWh exists a unified bill [54].

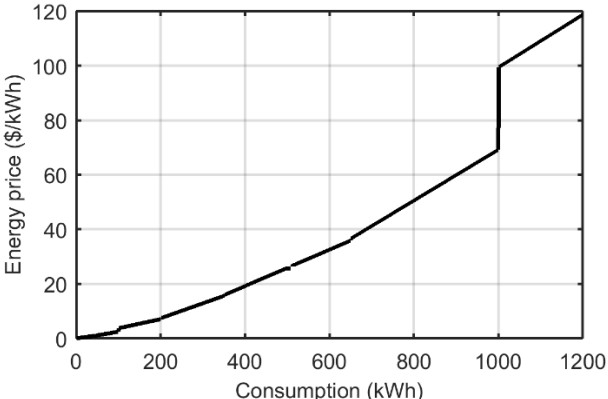

**Figure 7.** Energy consumption bill in Egypt.

The obtained results are compared to the GA and PSO optimizers to prove the developed CS optimizer's effectiveness. The developed optimizers are implemented via MATLAB 2015a. The interest rate, inflation rates, and technoeconomic analysis are given in Appendix B.

### 4.2.1. Scenario 1: Impact of Tilt Angle

To consider the impact of solar irradiation, historical data with an Excel sheet was recorded. The results in Figure 8 comprise several tilt angles. Keeping the solar PV at a horizontal axis is not a good option during energy harvesting. For lower angles, 0° to 30°, PV energy production takes a near-parabolic shape with smooth increases and decreases. For greater tilt angles, PV energy-gathering peaks at a fast rate. Beyond 70°, PV panel energy-gathering increases and decreases sharply. Provided that the standard meridian is 30°, investigating the load profile and the corresponding solar PV harvesting tilt angles from 40° to 70° might better meet the load demands, which peak at mid-day. Therefore, an average angle of 60° was chosen to meet load demands.

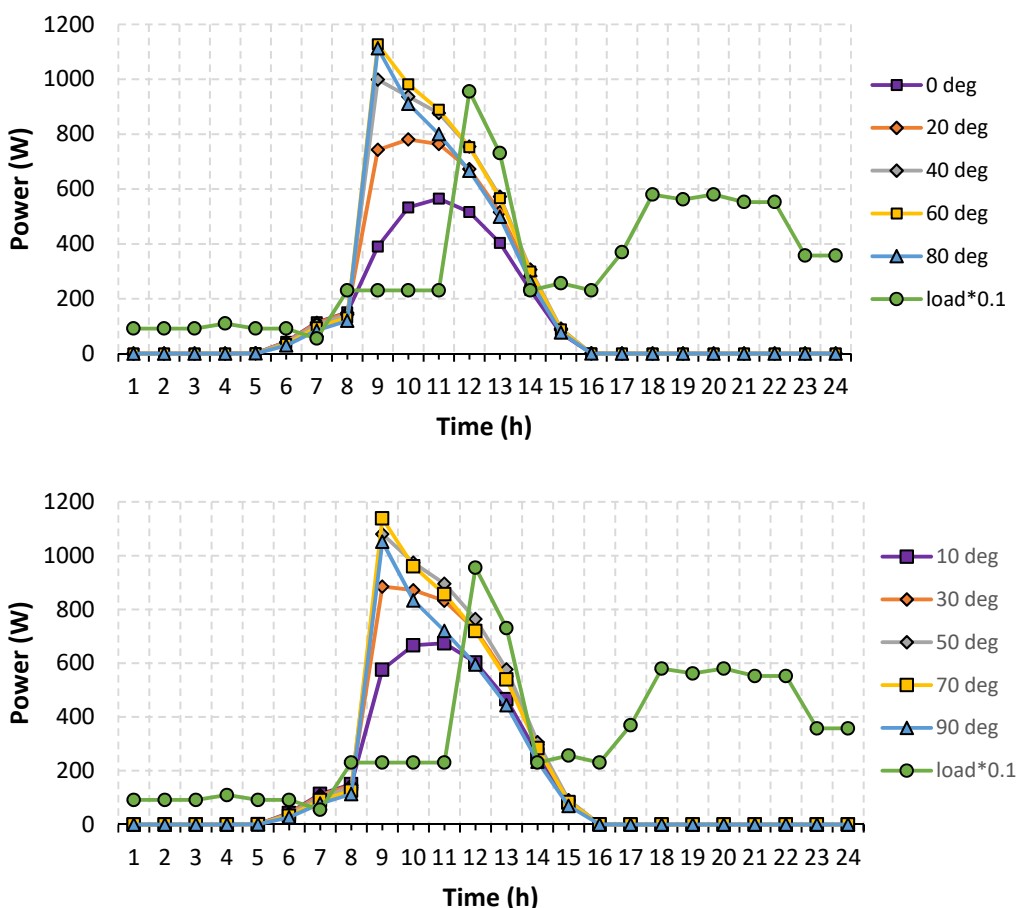

**Figure 8.** Tilt angle impact for solar power: upper the first group angles, and lower the second group.

Consequently, the corresponding global tilted solar irradiance is utilized in Equation (1) to help achieve the current technoeconomic study. The impact of the azimuth angle is demonstrated in Figure 9.

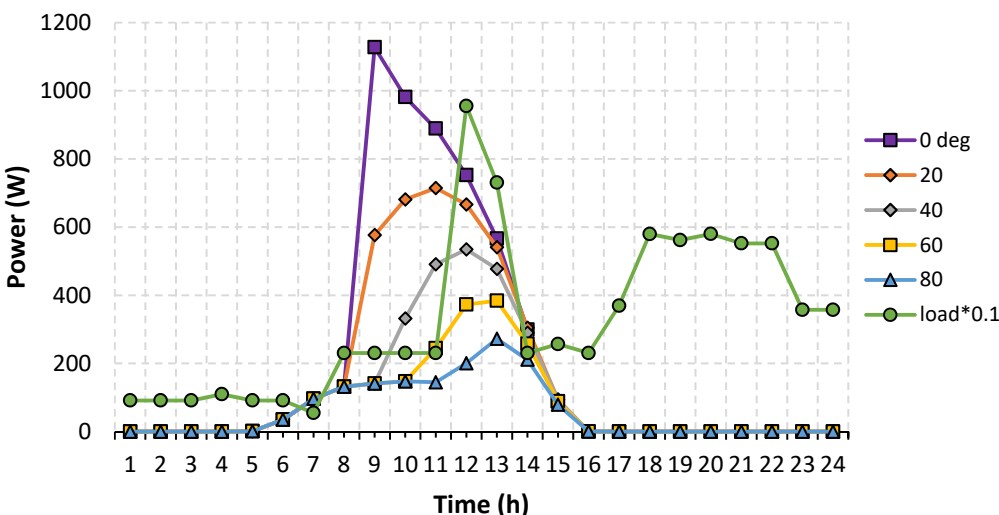

**Figure 9.** Tilt angle impact for solar power; azimuth angle at a 60° tilted angle.

While moving the module towards the south delays the global tilted irradiation peak, which satisfactorily meets peak demand, 40° is adapted to come across the load peak among the several considered azimuth angles.

### 4.2.2. Scenario 2: Economical Solution

In this scenario, the HMG system study demonstrated in Figure 2 aims to minimize total costs and delimit the optimal trading between the solar canal-top PV and the primary grid. The LCOE is identified as the objective function. The achievable results are shown in Table 2.

**Table 2.** The economic solution of canal-top PV.

| Algorithm | Canal-top PV (kW) | LCOE ($/kWh) | GHG (ton) | TNPC (k$) | LPSP | Yearly Evaporation * (m$^3$) |
|---|---|---|---|---|---|---|
| GA | 6.1 | 0.811 | 64.9 | 161.1 | 0 | 6.9 |
| PSO | 10.4 | 0.707 | 67.3 | 140.6 | 0 | 6.9 |
| CS | 10.4 | 0.707 | 67.3 | 140.6 | 0 | 6.9 |

\* Evaporation per 1 m length and 2 m wide channel.

The developed CS optimizer gives satisfactory results because it relaxes rapidly towards the optimal point, as in Figure 10.

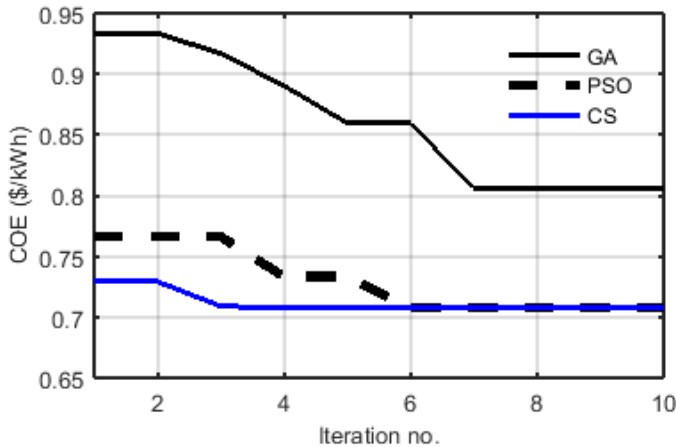

**Figure 10.** LCOE conversion for the developed optimizers.

Both the CS optimizer and PSO recorded the same LCOE. However, GA shows undesirably higher LCOE, which corresponds to an increase in the TNPC by 14.5%, referred to as the CS optimizer. The annual energy share is demonstrated in Figure 11, where the canal-top PV covers about half of load demand.

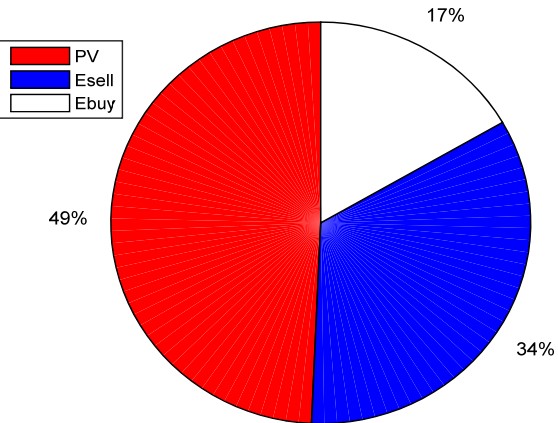

**Figure 11.** Annual energy share.

The economic feasibility study shows that a significant ratio is sold to the primary grid compared to the purchased energy. Evaporation is affected by several factors, such as wind speed, the channel's width, and ambient temperature. Evaporation estimation under the investigated weather conditions records 6.9 m$^3$/day for all optimizers.

### 4.2.3. Scenario 3: GIS Investigations

With the same results as in scenario 2, the canal-top PV modules' physical design via Helioscope online software was investigated to estimate the channel's required area [55]. Helioscope gives a chance to simulate the 3D design of the PV system by using the Google GIS and map system. The allocation process obtained via Google Maps was fed to the Helioscope software; accordingly, the required area is demonstrated in Figure 12.

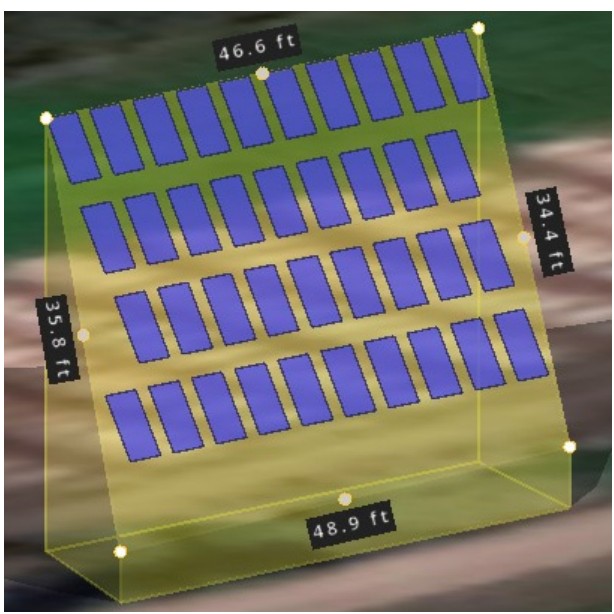

**Figure 12.** Helioscope top illustration accommodation investigation.

The PV is 13 ft (4 m) above the channel. Several simulation designs via Helioscope were carried out to determine an appropriate area of the near-optimal solution for the canal-top PV rating obtained in scenario 1 (i.e., 10.4 kW). It requires a 1630 ft$^2$ area of 35 ft average length with dimensions of 46.6, 34.4, 48.9, and 5.8 ft, respectively, to yield 10.6 kWp. Furthermore, Helioscope displays the system loss sources for the area under investigation, as in Figure 13.

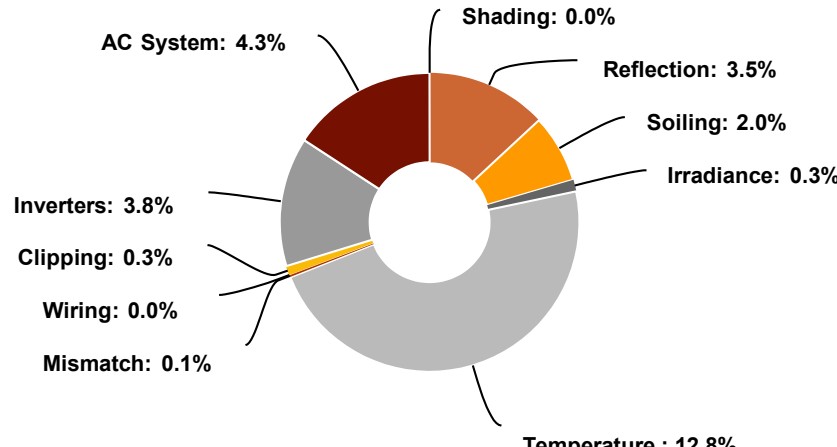

**Figure 13.** Canal-top PV loss via GIS investigation obtained through Helioscope simulations.

The investigation confirms that the ambient temperature has a significant impact on the PV operability. However, the system is safe, considering full or partial shading.

### 4.2.4. Scenario 4: Sensitivity of the Top Canal PV Accommodation

With the same conditions as in scenario 2, PV's impact upon the LCOE via the developed CS is considered in Figure 14.

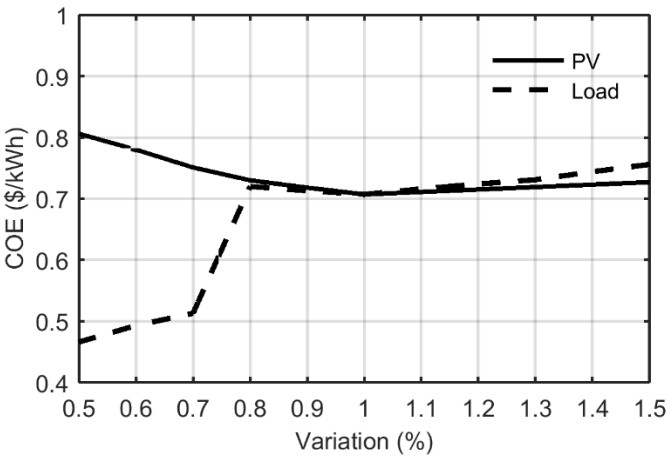

**Figure 14.** PV and load impact upon LCOE.

Away from the optimal point, canal-top PVs show an undesirable increase in the LCOE as the PV decreases. This behavior is because, as the PV decreases, the required energy is compensated from the grid. The prices increase in clusters according to the increased consumption to meet load requirements. For the same load demands, increasing the PV has a slight impact upon the LCOE, implying that drawing the required energy from the canal-top PVs is more favorable than purchasing it from the primary grid.

Contrarily, keeping the PVs fixed and decreasing the load significantly reduces the LCOE. This heavy drop comes from the nature of the price clustering demonstrated in Figure 7. Additionally, increasing the demand beyond the optimal point has a higher increase than the PVs' energy harvesting. To sum up, for a smaller load demand, purchasing the energy from the grid is favorable. However, for greater load demands, employing canal-top PVs outperforms the primary grid in the LCOE.

## 5. Discussions

From the above simulations and the case studies, the following discussion remarks can be drawn.

- The developed CS optimizer is validated through an impartial comparison with prior research, as demonstrated in Table 3. In [56], several battery-mix technologies were employed via a hybrid PSO-GOA algorithm. In [34], a bi-objective ant colony was conducted. Despite the figures in Table 3 depending on location, meteorological data, initial costs, microgrid configuration, salvage market, and the optimizer's ability to find a near-optimal solution, the developed CS seems to be competitive the other algorithms in the literature.

**Table 3.** CS effectiveness.

| Algorithm | LCOE ($/kWh) | GHG (ton) | TNPC (k$) |
|---|---|---|---|
| PSO$_{GOA}$ [56] | 0.658 | 141.8 | 118.8 |
| BOACA [34] | 1.082 | 118.3 | 121.6 |
| HOMER [34] | 0.809 | - | 30,033 |
| NGSA [37] | 0.19–0.25 | 8.87 | - |
| GWO [57] | 0.78–1.59 | - | - |
| Decision making | 0.12–0.17 | 100.3–130 | - |
| CS | 0.707 | 67.3 | 140.6 |

- According to the Helioscope investigation, the ambient temperature might significantly affect the canal-top PV accommodation performance, as in Figure 13. Contrarily, the soil in such agricultural areas has a negligible influence.
- The canal-top PV rating has a negligible influence compared to the load impact on the LCOE, as in Figure 14.
- The simulated results show that the PVs are superior to the main grid to meet the load demand, as illustrated in the annual energy share in Figure 11.
- The developed CS outperforms the other algorithms (Figure 10) as it relaxed fast towards the optimal solution.
- Water-saving through canal-top PV is a significant benefit; however, a quantitative analysis of the actual evaporation reduction and the quantity of the water saved will be part of future research.
- Another issue of the current work is related to the nature of the renewables, which could influence the utility grid frequency [58]. The modern controllers are to be designed to handle the canal-top PV injected energy and the main grid frequency oscillations.

## 6. Conclusions

The paper has tackled the power-sharing problem between a canal-top PV system and the primary grid to meet load demands. An innovative metaheuristic algorithm based on the CS algorithm was developed to solve the technoeconomic analysis. A mathematical formulation to calculate the evaporation over the canal to PVs was conducted. An estimation to obtain the area required over the canal is determined through the GIS algorithm via the Helioscope online software. Based on the above simulations and discussions, the following conclusions can be drawn: (1) The developed optimizer is generic and can be extended to solve many engineering problems; (2) According to the GIS investigation, the temperature at the investigated site has a substantial impact on the canal-top PVs' efficiency; (3) At greater demands, it is favorable to buy the energy from the canal-top PVs, while at lower loads purchasing the energy from the primary grid reduced the LCOE; (4) the proper tilt angle of the area under study is from 40 to 60 degrees with an azimuth angle 154° to the south; (5) increasing the canal-top PVs rating has less impact upon the LCOE. Meanwhile, it helps reduce the evaporation ratio with better environmental conditions. In our future research, further study on multiple renewable resource options is desirable and necessary to extend our knowledge of improving the economic and environmental conditions and conducting an in-depth analysis of evaporation reduction and electricity generation via a canal-top PV system.

**Author Contributions:** Conceptualization, A.A., A.K., J.J., and F.K.A.-E.; methodology, A.A., A.K., J.J., and F.K.A.-E.; software, F.K.A.-E., A.A., and A.K.; validation, A.A., A.K., J.J., and F.K.A.-E.; formal analysis, A.A., A.K., and F.K.A.-E.; investigation, A.A., A.K., J.J., and F.K.A.-E.; resources, A.A.; data curation, A.K. and F.K.A.-E.; writing—original draft preparation, A.A., A.K., J.J., and F.K.A.-E.; writing—review and editing, A.A., A.K., J.J., and F.K.A.-E.; visualization, A.K. and F.K.A.-E.; supervision, A.K. and J.J.; project administration, A.A. All authors have read and agreed to the published version of the manuscript.

**Funding:** This research received no external funding.

**Institutional Review Board Statement:** Not applicable.

**Informed Consent Statement:** Not applicable.

**Data Availability Statement:** Data are available whenever required.

**Conflicts of Interest:** The authors declare no conflict of interest.

## Nomenclature

| | |
|---|---|
| CS | Cuckoo Search |
| D | Saturation Deficit |
| DHI | Diffuse Horizontal Irradiation |
| E | Evaporation |
| GA | Genetic Algorithm |
| GHG | Greenhouse Gas Emissions |
| GHI | Global Horizontal Irradiation |
| GIS | Graphical Information System |
| HMG | Hybrid Microgrids |
| LCOE | Levelized Cost of Energy |
| LPSP | Loss of Power Supply Probability |
| NPC | Net Present Costs |
| PSO | Particle Swarm Optimization |
| PV | Photovoltaic |
| Rh | Relative humidity |
| SDGs | Sustainable Development Goals |
| Ta | Air temperature |
| V | Wind speed |

## Appendix A

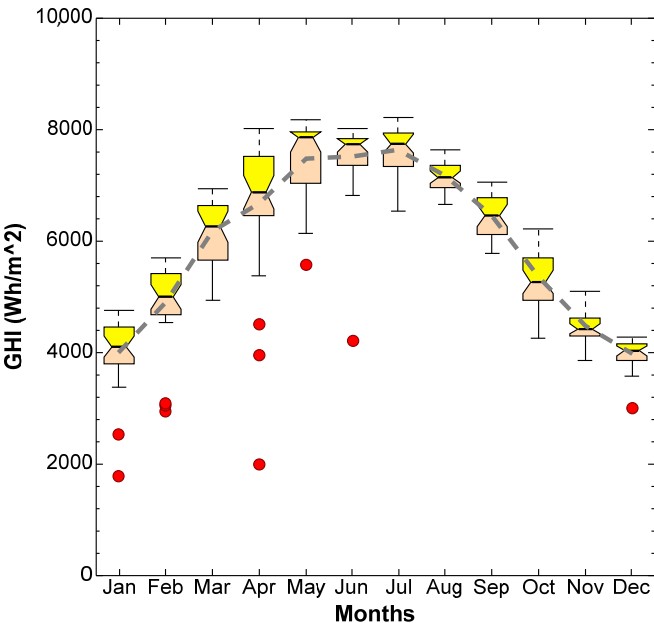

**Figure A1.** Notched boxplots show the daily variability of GHI for each month; the dashed line shows the mean values.

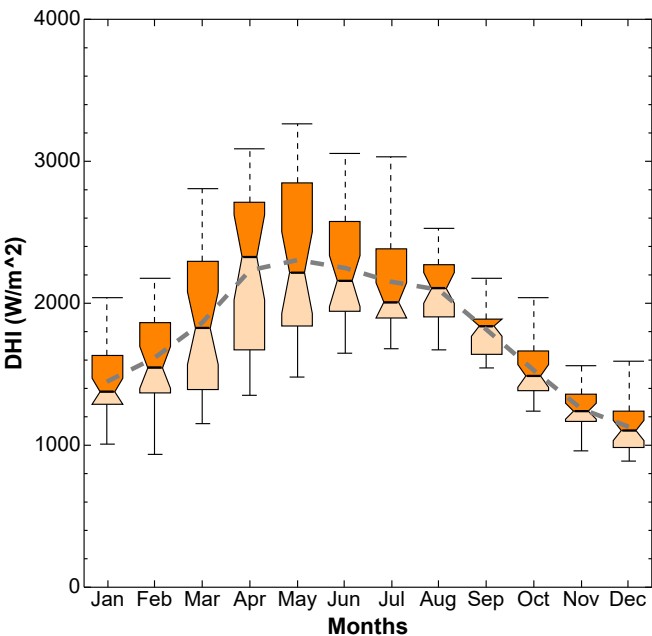

**Figure A2.** Notched boxplots show the daily variability of DHI for each month; the dashed line shows the mean values.

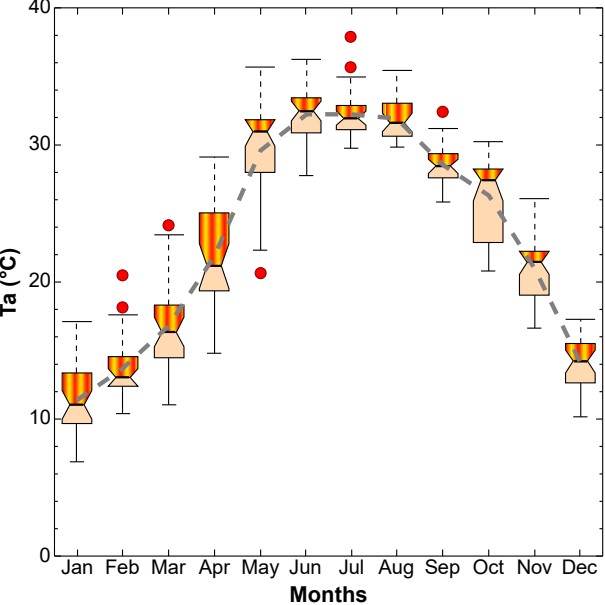

**Figure A3.** Notched boxplots show the daily variability of temperature for each month; the dashed line shows the mean values.



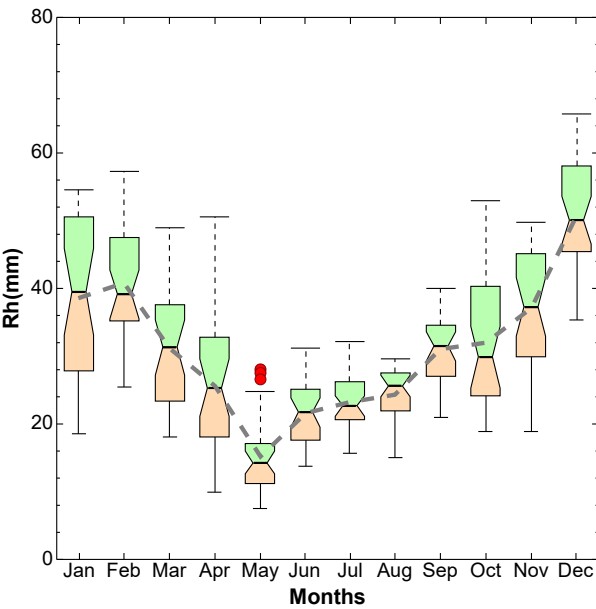

**Figure A4.** Notched boxplots show the daily variability of relative humidity for each month; the dashed line shows the mean values.

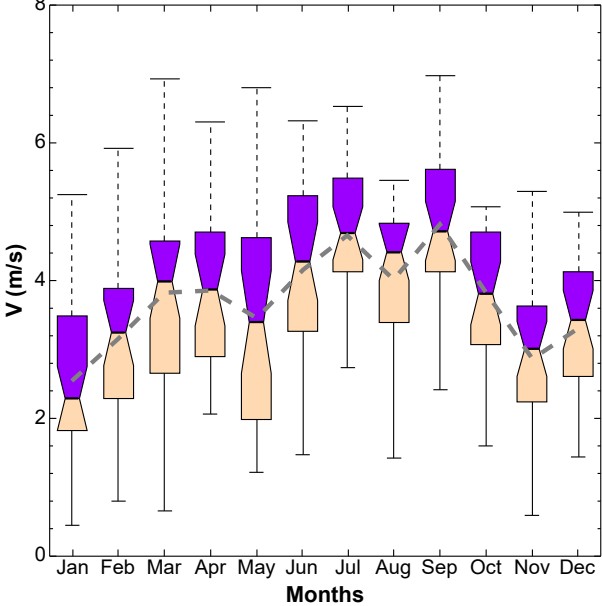

**Figure A5.** Notched boxplots show the daily variability of wind speed each month; the dashed line shows the mean values.

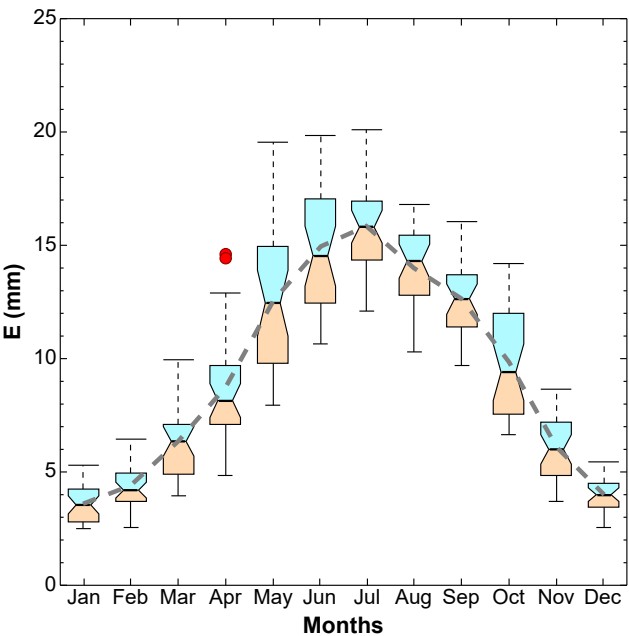

**Figure A6.** Notched boxplots show the daily variability of evaporation for each month; the dashed line shows the mean values.

**Appendix B**

- CS parameters: number of iterations = 10, number of host nests = 25, number of generations = 25, Pa = 0.25.
- PSO parameters: number of iterations = 10, population = 25, wo = 0.1, c1 = 0.25, c2 = 0.99, r1 = 0.3, r2 = 0.45.
- GA parameters: number of iterations = 10, population = 25, crossover probability = 0.6, crossover probability = 0.5.
- The technical and economic parameters of system compounds are given in Table A1.

**Table A1.** Technical and economic parameters of system components according to Abo-Elyousr and Elnozahy [34].

| Type | Value | Unit |
|---|---|---|
| **PV** | | |
| Lifetime | 20 | year |
| Initial cost | 600 | $/kW |
| Operational and maintenance cost | 0.01 | $/kW |
| CO2 emissions | 0.0225 | Kg/kWh |
| **Grid** | | |
| CO2 emissions | 0.143 | Kg/kWh |
| **Converter** | | |
| Lifetime | 20 | Years |
| Initial costs | 515 | $/kWh |
| Efficiency | 95 | % |
| **Others** | | |
| Project lifetime | 20 | Years |
| Interest rate | 13 | % |
| Inflation rate | 5 | % |

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
