# Peer review of "Energy Harvesting and Water Saving in Arid Regions via Solar PV Accommodation in Irrigation Canals"

_energies, doi:10.3390/en14092620_

Round 1

Reviewer 1 Report

The manuscript is interesting and it can contribute to improve the SDGs applied to irrigation systems.

Line 54. References should be discussed not only cite

Line 83-86. It should be justified

Line 99. The authors focused on Egypt, but they should do an analysis of PV installations  the rest of world farm or irrigation

Methods.

The authors studied the case study for 25 years but all conditions will be estable in this time? Do the authors the efficiency of the panels will be constant?.

Figure 8 should be discussed compared different angles

GA, PSO, and CS should be discussed and compared in Figure 9

An analysis of the applications of the SDGs should be developed since the authors described them in Figure 2. How do the authors think the SDGs are improved? Can it be quantified? Figure 5 and Figure 6 are not enough

The authors supposed sunny days but is not there cloudy days? How does it affect to the optimization results?

Author Response

Dear Reviewer, 

Thank you for your highly appreciated comments and remarks. The response file to your comments is attached  

Reviewer 2 Report

This manuscript entitled “Energy Harvesting and Water Saving in Arid Regions via Solar PV Power Accommodation in Irrigation Canals” is within the scope of the journal energies and the whole research presented is integrated, approaching various aspects of the application in a detailed way. It is satisfactory that authors take also into consideration the environmental impact of such solutions and they account for this specific case study (p.5,l.203-205) as, even the double benefit is incontrovertible, the potential impact on the ecosystems, e.g., due to the reduction of solar radiation to the water bodies, is a limitation in other cases. Finally, they discuss the evaporation reduction as an extra benefit that is also mentioned in Conclusions, however, the corresponding quantitative analysis is absent and, according to authors, it may be part of future research.

In general, the paper is well organized, well-written and concise, incorporating references’ information clearly into the text and citing everything in a proper way. The analysis is extensive and there are no comments regarding documentation. However, a slight language editing is necessary and some modifications are needed, thus, authors are advised to take into consideration the following suggestions/comments.

General comments:

Reference numbers are given as hyperlinks either in blue or in black – keep it constant.

Put some commas to make reading easier (e.g., “reducing water losses in irrigation networks, such as leachate and evaporation, might save remarkable amounts of water”) .

Minor mistakes:

In Abstract:

tech-no-economic --> techno-economic

In Introduction:

Spatio-temporal circulation (no uppercase is needed)

p.2,l.48 contributes: remove ‘s’

p.2,l.97 “The investigation … to investigate”

p.3,l.105 outdoors conditions: remove ‘s’ for outdoor

p.3,l.124 show the way forwards: remove ‘s’

p.3,l.127 to find a near-optimal solutions: remove ‘s’

p.3,l.141-151 please rewrite these sentences.

Fig.1.b: “Case Study” – also, rewrite the figure’s caption

In Section 2.5.2: Genatic Algorithm --> Genetic

In Section 3.4: Authors are advised to add a column in Table 1 with the E(mm) average monthly values.

Author Response

Dear Reviewer, 

Thank you for your highly appreciated comments and remarks. The response file to your comments is attached.

Best Regards  

Reviewer 3 Report

It is a very interesting work, although I have missed the validation of the results obtained, even with a scale model.

Author Response

(The authors gave the same response as above.)

Round 2

Reviewer 1 Report

The authors clarified all suggestions